# The Effect of Artificial Intelligence on End-User Online Purchasing Decisions: Toward an Integrated Conceptual Framework

**Hasan Beyari** [1,*] and **Hatem Garamoun** [2]

1  Department of Administrative and Financial Sciences, Applied College, Umm Al-Qura University, Makkah 24382, Saudi Arabia
2  Marketing Department, College of Business Administration, University of Business and Technology, Jeddah 21448, Saudi Arabia
*  Correspondence: hmbeyari@uqu.edu.sa

**Abstract:** This study was an investigation into the effect of selected artificial intelligence tools and the consideration set on the end-user purchasing intentions of convenient and shopping products of Saudi Arabian customers. The consideration set was the factor that the researcher sought to associate directly with the online purchasing intention variable. The selected AI tools and approaches were machine learning, purchase duration, social product recommendation, and social media dependency. The four served as the indirect factors, as their effect was measured against the consideration set variable. The theoretical framework employed in this study comprised the unified theory of acceptance and use of technology (UTAUT) and the theory of reasoned action. The researchers used an online survey with a sample of 148 customers. In analyzing the findings, the researchers opted for the structural equation modeling (SEM) approach. The findings indicated evidence of association with a consideration set of three independent variables, namely, machine learning, purchase duration, and product recommendation. The study also established that customer consideration sets influence end-user purchase decisions for online customers.

**Keywords:** artificial intelligence; machine learning; purchase duration; product recommendation; consideration set; social media dependency; end-user purchase decisions



## 1. Introduction

The ecommerce landscape is highly dynamic because of the technological sophistication evident in the industry. Whenever new technology arrives, many such businesses openly adopt it with the intention of improving their competitiveness. Artificial intelligence is one of the many technologies that ecommerce proprietors have embraced. According to [1], ecommerce was the leading adopter of artificial intelligence, and it was closely followed by the Fintech sector. For the past 5 years, its adoption has seen a significant increase. According to [1], 84% of these businesses are estimated to be either already embracing artificial intelligence or considering to do so. Moreover, ref. [2] suggests that 20% artificial intelligence has resulted in 20% additional revenue for ecommerce companies.

Almost every online shop has access to valuable client information that can be utilized to improve targeted marketing. Most firms, on the other hand, are unable to make use of the terabytes upon terabytes of data they have at their disposal. When it comes to big data analytics, artificial intelligence is used to automate the processing of enormous data sets, which is useful in this situation. Machine learning is being used by several major retailers to continuously improve their outcomes, which is a step above and beyond traditional marketing. The Luxury Escapes AI chatbot, for example, has raised the company's response rate on retargeting efforts by 89% [2], which is an example of how artificial intelligence may be utilized to improve marketing effectiveness. Numerous other firms, such as Lego, Subway, Esso, H&M, and Sephora, among others, have pursued similar initiatives with equally outstanding outcomes in recent years.

Artificial intelligence is playing a crucial role in converting interest into purchase intentions. A significant amount of the information that ecommerce businesses acquires pertains to future consumers or leads. AI can be used to reach out to both cold leads (who may not be familiar with the brand) and warm leads (who may be familiar with the brand) who have shown interest in the brand or product [1]. Apart from that, artificial intelligence is demonstrated to be a highly successful technique for remarketing clients. Retargeting is a type of marketing that is directed at buyers who are currently in the sales funnel but have not yet made a purchase decision. There are several approaches to artificial intelligence retargeting, but the majority of them contain customized business messaging. Conversational artificial intelligence (AI) is the most personalized type of commercial messaging available today.

Artificial intelligence has the potential to revolutionize the ecommerce industry in Saudi Arabia. This notion has been accepted by the many ecommerce platforms that have since adopted the technology and continue to improve upon it [3]. The ecommerce sector is global, and without complying with technological expectations, customers are likely to prefer foreign entities even when there are local firms in the same line of business. Projections indicate that transactions in the ecommerce sector will increase substantially in the next decade. Factors that have been observed to cause this increase in demand are a flourishing economy, Internet access, and high consumer purchasing power. This expectation has encouraged online shopping platforms to prepare technologically to increase their capacity and edge out their rivals in the market as they fight to expand their market share. The contribution of this research is to add new insights of AI tools to the knowledge of factors impacting end-user purchasing decision-making. This study also sheds light on areas of customer experience with online shopping in the context of AI.

The current paper comprises eight sections. The literature review section presents a critical analysis of the literature on AI tools, their application in marketing, the phenomenon of a purchase consideration set, and the impact of AI tools applied in marketing on both the purchase consideration set and end-user online purchasing decisions. The third section presents a theoretical background of the study and introduces the five research hypotheses tested in the study using structural equation modeling (SEM). The fourth section presents a research methodology used in the study. The fifth section analyzes the findings established in the study. The presentation and discussion of empirical data, conclusion, limitations, and future research can be found in the sixth, seventh, and eighth sections, respectively.

## 2. Literature Review

Scholarly interest in topics related to the effect of artificial intelligence on purchasing decisions and marketing success is growing, as these technologies become more prevalent in the business sector. The authors of [4] considered AI as a form of technological sophistication that often produces positive results for any business. The source was further supported by [5], where the researchers found that AI aids business decision making and customer retention strategies. The authors of [6] decried the many hurdles that managers must deal with as they integrate this technology into their businesses, especially in attracting and influencing customer purchase decisions. Nevertheless, the source firmly asserted that AI is becoming almost indispensable. The technology is primarily used on ecommerce websites to subtly influence customers toward buying more products or considering the purchase of specific additional items. According to [7], if left to the customers, they would be purchasing what is on their minds without caring for the array of items on sale. Online platforms are vastly different from conventional stores. One of the ways the two are different is that when checking out, a customer shopping at a physical store has visual access to other products that may trigger their interest [8]. Hence, online stores have to engage in heightened technological practices to simulate the features of physical stores and possibly enhance them.

In studying the effect of artificial intelligence on consumer behavior, [9] found that AI has the potential to transform the interactions customers have with the online platforms

into more productive ones. The authors of [10] also learned that AI makes the decision-making process more convenient for customers in their online interactions with ecommerce websites. Such decisions would be harder to make without the assistance of this technology. This notion is also evident in [11], where researchers indicated that the emergence of Industry 4.0 technologies have primarily centered on consumer behavior by improving metrics such as increasing customer impressions and purchase likelihood. One of the ways AI has positively contributed is through customer convenience and the ease with which such customers can access information on products sold on the platforms [5,12]. Access to information is not only desirable on the part of consumers but also by management. Having access to quality data and their analytics, management can make the proper decisions on segmenting the market and other critical customer relationship management issues [13]. This model maximizes the benefits accruing to the customer, as they receive value for their money by accessing the right products, and to management teams, as they can make customer decisions more optimally [14]. In this way, both customers and business managers appreciate the intrinsic value of artificial intelligence integration.

Machine learning (ML) is a specific technology in artificial intelligence that trains computers to predict human behavior by simulating it using specific statistical models. The authors of [15] found that ML is applied to the marketing field to predict customer preferences. By doing so, managers can customize their website to conform to the preferences expressed by customers and match their willingness to pay. The dynamic nature of machine learning is useful in creating different customer profiles, which is known as segmenting. These profiles provide a reliable basis for marketing specific products to a customer group. The authors of [16] also found the machine learning useful in influencing a user's consideration set by leveraging their demographic characteristics. The source argued that, by accurately segmenting the customer base, a firm is able to trigger customer interest in specific products with ease. Users' online activities and self-provided information are the sources of data used to establish which segment they should belong to [17,18]. The analysis of such voluminous data is almost impossible using other approaches. It is here that machine learning models help to capture the dynamics in customer preferences and product interest [19]. It is possible to understand a new customer's preferences by using data from other similar customers already shopping at an online store [20]. Rational customers seeking to maximize their utility are more likely to be influenced by marketing initiatives powered by machine learning than by generic and highly assumptive systems.

Purchase duration is the period between a user's first interaction with a product and the actual purchase. The authors of [21] defined it as the elapsed time between a consumer's first consideration of buying a product and the actual purchase itself. It is a well-known construct that has a significant influence on consumer buying behavior. Predictably, businesses prefer customers with the shortest purchase durations because the feedback is almost instant. Such customers are simpler to influence, especially using artificial intelligence systems. According to [22], purchase duration is a key variable that artificial intelligence systems in ecommerce use in predicting user behavior. The authors of [23] argues that purchase duration can also be viewed as a critical measure of the performance of an artificial intelligence system. A smaller purchase duration implies that a customer is highly influenced by the AI algorithm. Customers with a shorter purchase duration often do not put items on their carts, which is a list of items in their consideration sets. Carting significantly reduces purchase durations because one does not need to remember the name of the product [24]. Additionally, if one carts an item, it indicates to the system that the user is interested in such products. This information is critical, as the system may then send reminders to the customers [25]. Even without reminders, customers with carted items on their profiles are more active in closing purchase deals than those that cart less [26,27]. The consideration set is a useful metric that firms use to establish their customers' purchase intentions. The purchase duration's effect on the consideration set can boost online businesses seeking growth in sales.

Product recommendations are artificial intelligence systems that leverage the predictive power of sophisticated models to suggest products that are likely to influence a customer's interest. They are data-filtering systems that employ a variety of algorithms and data to offer the most pertinent products to a certain customer [28]. They do so by analyzing prior client activity (both current and historical), such as searches, clicks, and purchases. Then, they determine what will appeal to that consumer's preferences in the future [29]. AI-driven product recommendations assist clients in swiftly and simply locating things they wish to purchase [30]. Additionally, they enable firms to emphasize the things that other customers adore and to introduce those products to new consumers. Such systems, therefore, create the potential for cross-selling and upselling. The authors of [31] argued for the adoption of product recommendation AI systems by ecommerce businesses because evidence shows that such systems add value to the business. The source discovered that by recommending the most relevant products, businesses can soften customers into carting such suggestions. Product recommendations are also viewed by [32] as an essential tool for enhancing the quality of product suggestions. With such technology, a user views only what is most relevant to their situation, thereby spiking their interest. The authors [33,34] indicated that the more relevant the products a customer sees on their feeds, the better the chances of carting and ultimately purchasing them. It is entirely up to a user to decide whether they will cart the suggestions or put them in their 'favorites' list. In both cases, [35] found that it helps to put the product or products in a special place that is reachable anytime a customer needs to proceed with the purchase. The source also argued that product recommendation enhances a customer's commitment to a site because they can peruse and view products that they fancy.

Businesses and brands have leveraged social media to expand their reach. Businesses now have a better grasp of what their customers are saying and what they want. Management uses AI to research the most discussed topics and condenses them into trending lists to gain a better insight into consumer habits [36]. The effect of social media dependency extends beyond simple customer attraction. The initial stages of the purchasing decision process are characterized by heavy social media consultation [37,38]. These sites often have product reviews and other pertinent information that is critical to the purchase decision. This case manifests mostly when a high-value purchase is in the offing. Nevertheless, [39] found that even low-value product purchase decisions are partially influenced by the content that a customer has been seeing on their social media feed. The authors of [40] determined that brands and enterprises cannot ignore the impact of social media on consumer behavior. According to a Deloitte survey, consumers who are affected by social media are four times more likely to increase their spending on items [41,42]. Influencer marketing on social media is another avenue used by businesses to gain customer trust. By generating new clients, influencer marketing can perform wonders for a brand [43]. Numerous businesses are substituting YouTube, Instagram, and Snapchat influencers for celebrities. These online celebrities are providing shoppers with uncensored opinions on products that they adore. Additionally, the influence might be significant enough that 29% of buyers are more likely to make a purchase the same day they use social media [44]. These statistics indicate that the role played by social media dependency in forming a consideration set is significant.

Consideration set refers to the list of brands, whether physical or otherwise, that a consumer considers purchasing based on their need to fulfill a specific desire. Online platforms are unique in their potential to offer nearly limitless selections [45]. However, they frequently make assumptions about the purchasing process being frictionless, which may not always be accurate [46]. The authors of [47] investigated the effect of diverse search costs on the establishment of consideration sets and the response of consumers' shopping behavior to variations in the size of the online assortment. Their findings established that consideration sets greatly influence the purchasing decision of online consumers. The findings were echoed by [48], where the researchers argued that online purchase decision-making is heavily anchored to the carting behavior of digital customers. Although

some customers may cart products and then abandon this cart because they are financially incapable of proceeding with the purchase, ref. [49] revealed that a sizable proportion of them make the purchases anyway. The same study found that customers who frequently cart items are more active on ecommerce platforms. These findings were consistent with those established in [50], where carting behavior was compared to purchase behavior. The studies reviewed here seem to suggest that customer consideration sets tend to influence the sustainability of their purchasing behavior.

## 3. Theoretical Background

### 3.1. Theory of Acceptance and Use of Technology

The unified theory of acceptance and use of technology (UTAUT) is a model that explains the intentions of users as they interact with information systems and their implicit and explicit behaviors. The variables making up the model are effort expectancy, facilitating conditions, performance expectancy, and social influence [51]. Performance expectancy refers to the extent to which a user believes that using a system will help them achieve the desired level of performance. Effort expectancy is the amount of effort required to effectively use a system's features. Social expectancy is the degree to which a user believes that other people want them to use the system in question [51,52]. Facilitating conditions comprise factors that lead a user to believe that their organization has the necessary resources to support the system's usage. Although the four mentioned variables are the primary independent variables, the model includes some moderating variables, namely, gender, age, experience, and voluntariness of use [53]. However, not all the moderating variables mediate the effect of all relationships. For example, voluntariness of use mediates the effect of social influence on behavioral intention. The model was found to have an r squared coefficient of 0.7, which indicates that it explains 70% of the variation in the intentions expressed by users to work on a system. Figure 1 illustrates the causal and mediating effects of the model variables.

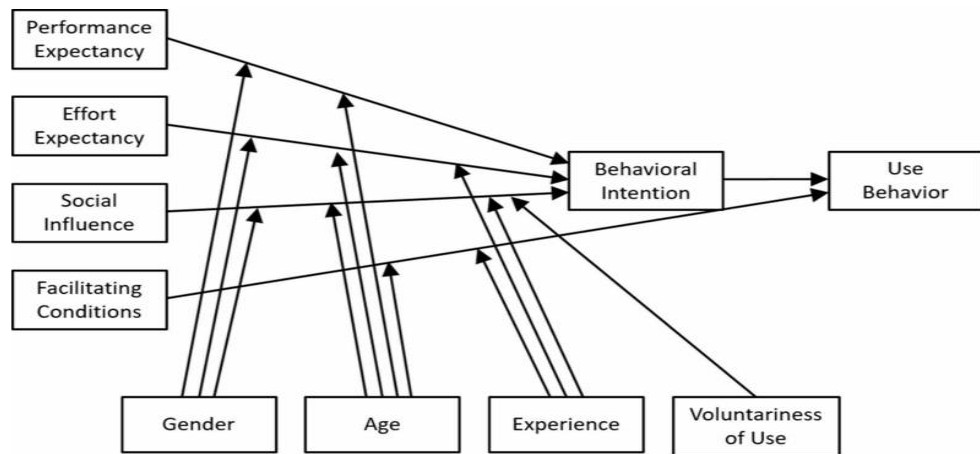

**Figure 1.** The UTAUT model [51].

### 3.2. Theory of Reasoned Action

The theory of reasoned action is another constituent of the study's theoretical framework. This concept explains the relationship between individuals' attitudes and behaviors. According to [54], there is a close link between behavioral intention and actual behavior. The latter, in turn, comprises such components as action, target, context, and time. Simultaneously, it should be noted that even a high behavioral intention does not always result in actual behavior, which is not addressed in detail by the authors of the TRA. According to [55], researchers and practitioners interested in predicting behaviors should examine antecedents of behavioral intentions. In their opinion, attitudes and subjective norms are the key factors affecting behavioral intentions, as shown in Figure 2.

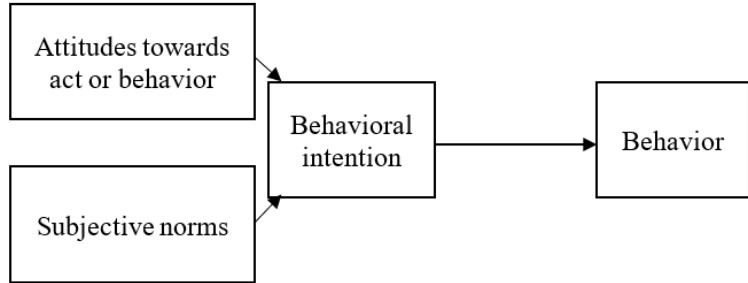

**Figure 2.** Theory of reasoned action [56].

*3.3. Conceptual Model*

The current study checked five hypotheses regarding the drivers of online end-user purchasing decisions. Figure 3 illustrates all the hypotheses that were examined in the research.

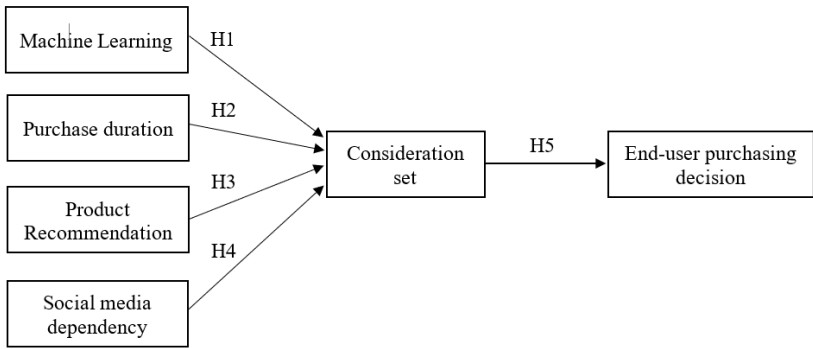

**Figure 3.** Conceptual framework of the study.

*3.4. Hypotheses Development*

3.4.1. Machine Learning on Customer's Product Consideration Set

Machine learning is a branch of artificial intelligence that many developers leverage because of its capacity to influence customer consideration. According to [57], machine learning models help businesses better understand the tastes and preferences of their customers. Additionally, these models register high accuracy scores when categorizing or segmenting customers based on their demographic data. They make it easy to fill landing pages with dynamic content that the customers are more likely to consider purchasing. Some of these items attract significant attention from other users because, based on this feature, other prospective customers are bound to consider them as potential purchases in the future. Machine learning models attentively read customers' purchase patterns and sometimes use common external databases or big data to determine the most effective ways of advertising to customers that trigger their interests. Even if the customer fails to make an instant purchase, the fact that they put the item in their initial or active consideration set is sufficient for a marketer. For this reason, the researcher hypothesized that:

**Hypothesis 1 (H1).** *Machine learning has a significant positive effect on a customer's product consideration set.*

3.4.2. Purchase Duration on Customer's Product Consideration Set

For the purposes of this study, the purchase duration variable refers to the amount of time that elapses between a consumer's first thought of buying a product and when they make the actual purchase. According to [25], a longer purchase duration results in a longer consideration set. The findings expressed in the cited study seem logical because it makes sense that a customer taking longer to purchase will have more brands to consider. On the other hand, a customer with a shorter purchase duration is not likely

to have so many purchase options. Businesses prefer customers with shorter purchase durations because such customers are easier to convince, especially with the use of artificial intelligence. In determining the length of a customer's consideration set, ecommerce platforms examine their carting and favoring behavior. Those wishing to make bigger purchase decisions tend to fill their carts and favorites pages with more products [58]. This tendency manifests because of the many dimensions that such high-value purchases have. Customers with longer purchase durations are more likely to be angling for a major purchase, and as such, prefer to fill their carts with several brands. For this reason, the researcher hypothesized that:

**Hypothesis 2 (H2).** *Purchase duration has a significant positive effect on the number of items on a customer's product consideration set.*

### 3.4.3. Product Recommendation on Customer's Product Consideration Set

Product recommendation is an inherent component of artificial intelligence applications in the ecommerce sector. The system works by evaluating a user, and based on their demographics and/or purchase history, it recommends the most relevant products to them. The engine is applicable in almost all situations, with item–item collaborative filtering being the most preferred approach. Systems adopting this approach examine the correlations between products and their user ratings [59]. This information is then compared to what other similar users have purchased. The ultimate suggestions are the items that a user has not rated or bought but that have been highly rated by similar users. Product recommendation is the basis of ecommerce because, in its absence, customers would only see products that are too general and irrelevant to their specific needs. Whenever one sees a highly relevant recommended item on the platform, they are likely to put it on their wish list. Hence, the researcher hypothesized that:

**Hypothesis 3 (H3).** *Product recommendation has a significant positive effect on the number of items on a customer's product consideration set.*

### 3.4.4. Social Media Dependency on Customer's Product Consideration Set

Sometimes, waiting for customers to come to an ecommerce platform may be an ineffective marketing technique. It becomes necessary to market firm products in spaces enjoying massive traffic of prospective customers, such as social media platforms. According to [60], advertising on social media effectively delivers marketing messages to the populace. Firstly, the platforms have guaranteed viewership because of their ever-busy pages. Secondly, one can select the audience to which the business wishes to send the marketing message. Thirdly, direct links to the main business website make it convenient for prospective consumers to access the product. Regarding their effect on the consideration set, social media sites allow users to interact with posts in various ways. In this case, a user can like, share, comment, or follow a post to receive future notifications on the same. Similarly, the user can save the post for future actions. For this reason, the relevant hypothesis states that:

**Hypothesis 4 (H4).** *Social media dependency has a significant positive effect on the number of items on a customer's product consideration set.*

### 3.4.5. Customer Consideration Set on the End-Use Online Purchasing Decision

A consideration set is the number of items on a customer's wish list. Regardless of the length of this list, a consumer may choose to proceed with the purchase or not. According to [61], people who favorite items on ecommerce platforms usually end up purchasing the products in the future. Whenever one puts an item on a wish list, it is because its features have impressed them. This list barely changes unless the item is out of stock. Nevertheless, whenever they navigate to the 'favorites', 'wish list', or 'cart' page, the sight of the products will remind them of their previous intention to purchase. The

result is an urge to complete the purchase and acquire the listed products, and this option is always conveniently available for the customer to use. If they are financially capable of making the purchase, many customers find the urge irresistible. Therefore, the researcher hypothesized that:

**Hypothesis 5 (H5).** *Customer consideration set has a significant positive effect on the end-user online purchasing decision.*

## 4. Research Methodology

The study considered a descriptive survey research design, which was inspired by the positivist research philosophy adopted throughout the paper. Using this design, the researcher established that they would collect data at customer level. Nevertheless, ecommerce developers were spared the technical questions since they have the right level of competence to respond to them. The voluminous number of potential respondents presented a challenge to the researcher. However, the researcher contacted one ecommerce platform (United Electronics Company—extra.com) (accessed on 18 March 2022) operating in Riyadh, Saudi Arabia. In minimizing the potential number of respondents and maximizing the quality of responses, the researcher reached out to the target ecommerce platform via its Facebook page. The requirements for inclusion were users who had contacted the company via a direct message in the last 2 weeks on business-related issues. Sending a direct message to a page is one of the high-level interactions a social media user can have. It was established that the total number of users fitting this profile as 209, and they formed the basis of the study population. The researcher relied on simple random sampling where all members had an equal chance of participating.

### 4.1. Item Measurement and Questionnaire Design

A quantitative research methodology was utilized in the research. Considering that qualitative techniques cannot be used to quantify variables, such a choice was natural [62]. The researcher utilized the online survey method to collect quantitative data. An online survey is one of the most popular research methods simultaneously characterized by efficiency, simplicity, and effectiveness [63]. The main questionnaire included AI-based product recommendations, purchase duration, social media dependency, consideration set, and the end-user purchase decision, as seen in Table A1, Appendix A. However, a side-questionnaire to collect demographic information was also developed. The items were quantified with the help of Likert's 5-point scale where: 1 = strongly disagree (SD); 2 = Disagree (D); 3 = neither Agree nor Disagree (N); 4 = Agree (A); 5 = Strongly Agree (SA). Appendix A shows the survey questions used in the study.

### 4.2. Sampling and Data Collection

A sample is a subset of the total population under investigation. The study population was 209, which was the number of participants fitting the requirements. Using Yamane's formula, the researcher determined that the minimum sample size was 137. This number formed the basis of the analysis. The researcher used simple random sampling to select who among the 209 potential respondents was to participate. It was a probabilistic sampling technique that gave all members a chance to participate in a survey [64]. It was deemed suitable because all the respondents were similar in their characteristics. The only prerequisites for admission were that one should have made an online purchase in the past 3 months and was willing to participate in the study. A post was made on Facebook, which attracted the 209 members specified above. Those sampled were emailed questionnaire links, which they had shared while indicating their interest.

The questionnaire was hosted on Google Forms, an online platform developed and owned by Google to facilitate data collection for research. All questions were posted on the site, and respondents followed a link to access the questions and participate. The researcher used simple English to design the questionnaire to make it convenient, even for members who may not be proficient in English. Furthermore, it was translated into

Arabic to cater to those with little to no English knowledge. The questionnaire was sent to 155 respondents. The outcome was encouraging, as 148 participants promptly completed the questionnaire, thereby suggesting a 95.4% response rate. Better still, no question was left unanswered. The researcher facilitated this outcome and checked on who had not answered their questionnaire, sending them reminders. Those who had partially filled it were also reminded.

*4.3. Structural Equation Model*

The paper adopted the structural equation modeling approach in its analysis. The path diagram representing this modeling approach is shown in Figure 4.

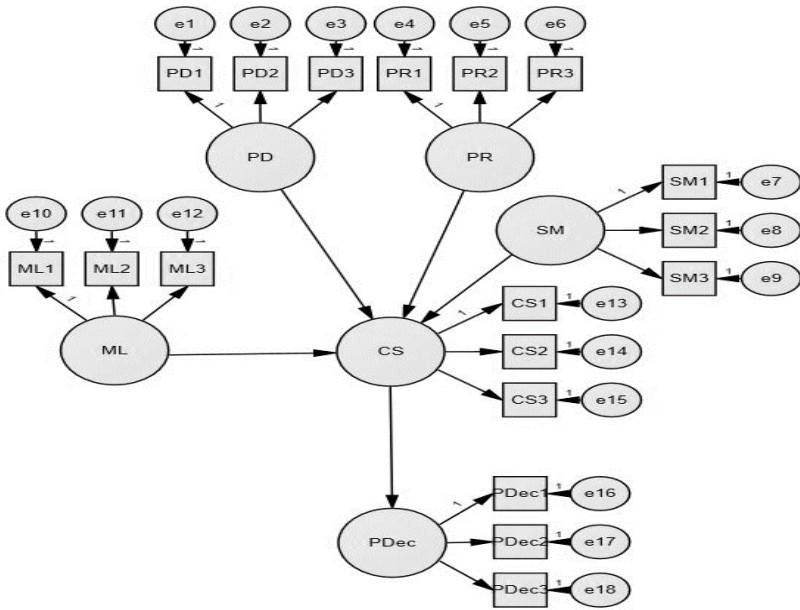

**Figure 4.** Path diagram for the structural model.

## 5. Data Analysis

*5.1. Analysis of Respondents' Profiles*

5.1.1. Distribution of Respondents by Gender

The first parameter of interest was gender. Here, the goal was to establish the distribution of respondents based on whether they were male or female. Findings indicated that female respondents dominated the sample, as they were 58.8%, and their male counterparts comprised 41.2%. Table 1 illustrates the distribution of respondents with respect to their gender affiliations.

**Table 1.** Demographic characteristics.

| Demographics | Frequency | Percentage (%) |
|---|---|---|
| Gender | | |
| Male | 61 | 41.2 |
| Female | 87 | 58.8 |
| Age | | |
| 18–25 | 29 | 20 |
| 26–30 | 50 | 34 |
| 31–40 | 47 | 32 |
| 41–50 | 17 | 11 |
| >50 | 5 | 3 |

### 5.1.2. Distribution of Respondents by Age

The second parameter of interest was the age characteristics of respondents. Findings revealed that the biggest age group was members between 26 and 30 years, as it constituted 34%. The group was closely followed by respondents of 31–40 years, a group whose composition was 32%. The least populated group was members over 50 years, as they were 3%. Table 1 illustrates the distribution of respondents according to their ages.

### 5.2. *Assessment of the Measurement Model*

Convergent and discriminant validity measures the validity of the questions asked to the respondents. This assessment helped to establish whether the responses would help attain the study's objectives. Indicator reliability and internal consistency measured the reliability of the instrument. These measures helped establish the extent to which the study could be replicated and still deliver similar outcomes.

### 5.2.1. Indicator Reliability

This metric measures the degree of similarity in the variation of question variables relative to their target variables. This measurement is critical because it indicates whether the question variables contribute to the overall variance of their respective target variables. According to [65], the minimum threshold for acceptance is 0.7. The analysis conducted in this study revealed that all the construct variables scored values more significant than this required minimum value. The average of the composite reliability score was 0.80. Table 2 shows the results obtained from running the analysis in AMOS.

**Table 2.** Construct indicators.

| Construct | Items | Factor Loading | Composite Reliability | Indicators | Cronbach's Alpha | AVE |
|---|---|---|---|---|---|---|
| Machine learning | ML 1<br>ML 2<br>ML 3 | 0.846<br>0.563<br>0.724 | 0.76 | 3 | 0.895 | 0.720 |
| Purchase duration | PD 1<br>PD 2<br>PD 3 | 0.825<br>0.662<br>0.708 | 0.78 | 3 | 0.764 | 0.735 |
| Product recommendation | PR 1<br>PR 2<br>PR 3 | 0.835<br>0.684<br>0.869 | 0.84 | 3 | 0.784 | 0.800 |
| Social media dependency | SMD 1<br>SMD 2<br>SMD 3 | 0.91<br>0.773<br>0.902 | 0.90 | 3 | 0.831 | 0.864 |
| Consideration set | CS 1<br>CS 2<br>CS 3 | 0.713<br>0.751<br>0.771 | 0.79 | 3 | 0.871 | 0.745 |
| End-user purchase decision | EUPDC 1<br>EUPDC 2<br>EUPDC 3 | 0.659<br>0.665<br>0.716 | 0.72 | 3 | 0.827 | 0.680 |

### 5.2.2. Internal Consistency

The internal consistency metric measures how constituent variables explained the latent variables assigned to them. This reliability metric is best measured using Cronbach's alpha, conveniently processed in SPSS. With indicator or composite reliability, the minimum expected value is 0.7 [65]. The findings suggested that all variables scored an internal consistency value more significant than the required minimum value. The average score was 0.829 as shown in Table 2.

### 5.2.3. Convergent Validity

Convergent validity measures the relationship between exogenous variables and their parent variables. The measure is critical in establishing how these two sets of variables correlate. A higher correlation is desirable, and it is measured using the Average Variance Extracted (AVE). The minimum required value is 0.6 [66], and all the variables under investigation met this requirement. The machine learning scored the lowest AVE (0.720), whereas social media dependency scored the highest (0.864). Table 2 shows the computation procedure and the ultimate AVE scores for the variables under investigation.

### 5.2.4. Discriminant Validity

Discriminant validity is a validity measure that appraises whether or not questionnaire items explain their target variable more than they explain other non-target variables. It is logical to expect responses under variable A to correlate with variable A more than they correlate with variable B. The rule of thumb is that the correlation coefficient between a variable's questionnaire items should be higher than they correlate with others. Based on this criterion, the instrument was valid. Table 3 shows the results computed from the output obtained from the AMOS software.

**Table 3.** Discriminant validity results.

|  | **ML** | **PD** | **PR** | **SMD** | **CS** | **EUPD** |
|---|---|---|---|---|---|---|
| ML | 0.7916 | | | | | |
| PD | 0.5066 | 0.8282 | | | | |
| PR | 0.2630 | 0.2880 | 0.8968 | | | |
| SMD | 0.0207 | 0.0229 | 0.0000 | 0.8544 | | |
| CS | 0.2447 | 0.2680 | 0.0000 | 0.0000 | 0.8544 | |
| EUPD | 0.3600 | 0.3947 | 0.0000 | 0.0000 | 0.0000 | 0.7998 |

### 5.3. Assessment of the Structural Model

Findings from the structural model indicated that the machine learning, purchase duration, and product recommendation significantly influenced the consideration set, as they scored beta values greater than 0.22. However, the social media variable failed to achieve this minimum value, as it scored a beta of 0.02. The effect of the customer consideration set on their purchasing decisions was affirmed by the beta of 0.97. Figure 5 shows the results of the structural model obtained from AMOS.

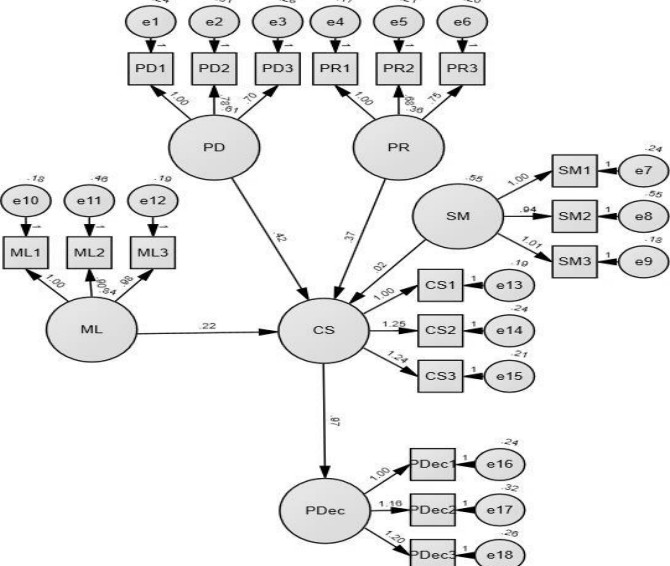

**Figure 5.** Results of the structural model.

### 5.3.1. Fit Indices for the Structural Model

The structural equation model scored a chi-square coefficient of 497.976 (df = 124, $p$ = 0.000) and a CMIN/DF score of 4.016. According to [67], if the CMIN/DF score is below 5.0, it indicates a reasonable fit. This was an indication that the model was statistically significant. Table 4 shows that the default, saturated, and independence models all attested to the same conclusion.

**Table 4.** Fit indices for the structural model.

| Model | CMIN | DF | $p$ | CMIN/DF |
|---|---|---|---|---|
| Default model | 497.976 | 124 | 0.000 | 4.016 |
| Saturated model | 0.000 | 0 | | |
| Independence model | 2536.617 | 153 | 0.000 | 16.579 |

CMIN = Chi-square value; DF = Degree of Freedom.

### 5.3.2. Regressing Consideration Set on Machine Learning, Purchase Duration, Product Recommendation and Social Media Dependency

The multiple regression analysis conducted on the data suggested that the overall r squared coefficient for the model was 0.796. It implies that 79.6% of the variation in the consideration set can be explained by the variation in the selected factors (machine learning, purchase duration, product recommendation, and social media dependency). The model was statistically significant from the F statistic score of 139.229 (df = 147, $p$ = 0.000). Regarding the coefficients, all AI construct variables significantly influenced the consideration set, except social media dependency. The said variable (social media dependency) scored a beta of 0.017 (t = 0.422, $p$ = 0.673). The most significant AI factor was machine learning, which scored a beta of 0.437 (6.618, $p$ = 0.000), as shown in Table 5.

**Table 5.** Regression of CS on ML, PD, PR, and SMD.

| | | Summary Stats | | | |
|---|---|---|---|---|---|
| **R Squared** | **df** | **Adj. R Squared** | **F Score** | **Sig/$p$-Value** | |
| 0.796 | 147 | 0.790 | 139.229 | 0.000 | |
| | | **Coefficients** | | | |
| **Variable** | | **Beta** | **t-stat** | **$p$-Value** | **Decision** |
| ML | | 0.437 | 6.618 | 0.000 | Supported |
| PD | | 0.275 | 4.903 | 0.000 | Supported |
| PR | | 0.274 | 4.841 | 0.000 | Supported |
| SM | | 0.017 | 0.422 | 0.673 | Rejected |

### 5.3.3. Regressing End-User Purchase Decision on Consideration Set

Findings from this subsection established that the customer consideration set significantly impacted their purchase decisions. The r squared coefficient was 0.920, which means that 92.0% of the variation in the end-user purchase decision variable can be explained by the variation in a customer's consideration set. The model scored an F statistic of 1687.75 (df = 147, $p$ = 0.000). Table 6 summarizes the results obtained from SPSS. The consideration set variable scored a beta of 0.959 (t = 41.082, $p$ = 0.000).

**Table 6.** Regressing purchase behavior on consideration set.

| | | Summary Stats | | | |
|---|---|---|---|---|---|
| **R Squared** | **df** | **Adj. R Squared** | **F Score** | **Sig/$p$-Value** | |
| 0.920 | 147 | 0.920 | 1687.752 | 0.000 | |
| | | **Coefficients** | | | |
| **Variable** | | **Beta** | **t-stat** | **$p$-Value** | **Decision** |
| CS | | 0.959 | 41.082 | 0.000 | Supported |

## 6. Discussion

The study determined that machine learning is a very effective marketing instrument that can help companies integrate their products and services into the consideration set of potential consumers. The same conclusion was arrived at in the work of [16]. It is an indication that, as firms adopt machine learning in their marketing approaches, they stand a better chance of having customers show more interest in their brands. This notion is evident in [15], where the researchers found that machine learning adapts to customer preferences and can leverage this power to influence them. As expressed in [24], if a user adds an item to his or her cart, the system will know that the user is interested in that particular product. This information is essential since the system may use it to send reminders to the clients. Customers who have cart items on their profiles are more active in closing purchase deals than customers who have fewer cart items on their profiles, even if they do not receive reminders. In order to better understand their customers' purchasing intentions, businesses can use the consideration set as an effective measurement tool. By investing in building AI infrastructure aiming at identifying, analyzing, and interpreting the variables affecting customer decisions, this research argues that firms should utilize machine learning and social media insights in improving performance and gaining competitive edge.

Product recommendation has also been observed to impact how a customer populates their consideration set significantly. These findings are consistent with those established in [59], where the researchers found that AI-based product recommendations significantly impact customer considerations to purchase items. The research results are aligned with the UTAUT model by illustrating an evident connection between behavioral intention and use behavior. Furthermore, the fact that machine learning, product recommendation, and purchase duration were found to influence customers' consideration sets could be interpreted as evidence that the three variables are critical in molding consumer behavior. The fact that the consideration set significantly impacts end-user purchase decisions conforms with the theory of reasoned action. As the authors of [48] reported, the consideration set is an integral element in customers' decision-making processes. Items to which customers attach enormous weights in the initial and active consideration sets have high purchasing chances.

Interestingly, this study found that social media dependency has a positive but insignificant effect on the consumer consideration sets. This revelation contrasts to [39], which indicated that social media dependency is critical in influencing consumer consideration sets and their ultimate purchase decisions. A closer examination of the factor reveals it to be the odd one out among the four selected factors. Social media dependency is a factor that is mainly exercised outside the ecommerce systems since third-party companies host the platforms. Hence, firms do not have as much control over the platforms' mechanics. Therefore, it is plausible that the other internally managed factors such as machine learning, product recommendation, and purchase duration rank higher in their influence on consideration sets.

## 7. Conclusions

The research findings confirmed that an understanding of the customer consideration set is critical for achieving desired changes in their online purchasing behavior. AI tools seem to be an effective instrument for achieving this goal because they provide valuable data on various aspects affecting customers' behavioral intentions. They also provide effective instruments that impact end-user behaviors by promoting convenient or shopping product recommendations and customizing product offerings. They are also responsible for promoting other measures that increase the likelihood of a sustainable online purchase. This is especially the case if facilitating conditions, such as the purchase duration and social media dependency, are favorable for marketers. Internally managed factors, namely, machine learning, product recommendation, and purchase duration, significantly influenced customer consideration sets. These findings imply that ecommerce businesses should leverage these factors in their efforts to increase customer attention to products sold by a

company. Although social media is not a significant predictor of consideration sets, it is still a great marketing channel that effectively enhances other aspects of a business.

The study has a couple of significant limitations that might limit the applicability of its findings. First, the research relied on reported data from consumers, even though some of these responses might not have been entirely accurate. Second, the construct of online purchase decisions was examined in the research through the prism of a purchase intention, even though a high purchase intention might not always translate into an actual purchase.

There are several areas for further research that seem especially promising based on the research findings. First, scholars could consider examining the mediation effect of demographic constructs on a relationship between various independent variables and the customer consideration set. The UTAUT model states that the age, gender, experience, and voluntariness of use might strongly influence how various constructs affect users' behavioral intentions. Second, it might also be promising to explore potential constructs that could moderate the relationship between the customer consideration set and online purchasing behavior.

**Author Contributions:** Conceptualization, H.B.; Data curation, H.G.; Formal analysis, H.B. and H.G.; Funding acquisition, H.G.; Investigation, H.G.; Methodology, H.B.; Project administration, H.G.; Resources, H.B.; Software, H.G.; Validation, H.B. and H.G.; Visualization, H.B.; Writing—original draft, H.G.; Writing—review & editing, H.B. All authors have read and agreed to the published version of the manuscript.

**Funding:** This research received no external funding.

**Institutional Review Board Statement:** Not applicable.

**Informed Consent Statement:** Not applicable.

**Data Availability Statement:** The data presented in this study are available on request from the corresponding author.

**Conflicts of Interest:** The authors declare no conflict of interest.

## Appendix A

**Table A1.** The Questions Used in the Survey.

| Construct | Questions | Inspiring Source |
|---|---|---|
| Machine learning | Ecommerce systems significantly rely on classification techniques such as KNN and naïve Bayes to classify users and products | [57] |
| | Ecommerce systems significantly embrace regression techniques such as logistic regression and decision trees to make predictions | [57] |
| | Ecommerce systems significantly leverage clustering techniques such as k-means and neural networks to group phenomena based on shared features | [57] |
| Purchase duration | You take long before you can purchase the product you have been planning to acquire | [58] |
| | You do not make rushed decisions when considering whether to purchase products online | [58] |
| | You have several considerations to make before purchasing products from ecommerce websites | [58] |
| Product recommendation | Ecommerce websites recommend relevant products to you | [59] |
| | Product recommendations made to you are based on your previous purchases | [59] |
| | You have purchased at least some of the products recommended to you | [59] |
| Social media dependency | My social media feed sometimes features marketing messages from ecommerce companies | [60] |
| | I follow links to ecommerce websites from my social media accounts | [60] |
| | I follow several ecommerce companies on my social media handles | [60] |
| Consideration set | You have several products on your wish list on ecommerce websites | [61] |
| | Your wish list keeps expanding as you interact more with ecommerce websites | [61] |
| | Each time you visit these sites, you find an item that intrigues you to the point of considering buying it in the future | [61] |
| End-user purchasing decision | You are likely to purchase items you like from ecommerce websites | [68] |
| | You have engaged in repeat purchases while interacting with ecommerce websites | [68] |
| | Making the decision to buy is aided by features embedded in ecommerce websites | [68] |

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
