# Peer review of "The Effect of Artificial Intelligence on End-User Online Purchasing Decisions: Toward an Integrated Conceptual Framework"

_sustainability, doi:10.3390/su14159637_

Round 1
Reviewer 1 Report
The paper lacks a clear conceptual framework linking the extant literature and what the authors are intending to study in current research. Actually, I cannot see how the two theories (UTAUT and TRA) are associated with the development of the conceptual model in Figure 3. The conceptual model is not self-explanatory. Elaborations are needed.
Moreover, the conceptual model looks problematic. The use of machine learning affects the purchase duration and product recommendation. Therefore, purchase duration and product recommendation. In fact, it is inappropriate for the authors to classify "purchase duration" and "product recommendation" as "selected AI tools and approaches".
Regarding the research methodology, I doubt how a "random sample" was really achieved. The sampling strategy needs further elaborations and explanations.
What are the meanings of 5 and 1 for the Likert scale used in survey?
How did the authors ensure wordings of the translated version of the questionnaire shared the same meanings as those in the original version?
The meanings of the constructs are unclear? For example, what does "machine learning" means in the conceptual model? I don't think the three questions in Appendix A really measures "machine learning".
The presentation of the research findings is also weird. The authors should be careful in dealing with the statistics. For example, the frequencies and percentages of the survey respondents in terms of gender are not matching in Table 2.
Was the type of product considered or controlled in the research? That factor was evidenced to be a significant determinant of the online purchase behaviour of people.
Last but not least, the paper has a lot of typos and grammatical errors. The authors should have the paper proofread by a professional English writer before submission.
Author Response
Dear reviewer 1, Please see attached file.

Reviewer 2 Report
This paper reports the results of a case study in Saudi Arabia on the impact of artificial intelligence tools' assistive features on customers' purchasing decisions.
Although the sample size of 148 people is trivially small, the results of a careful study of consumer behavior in Saudi Arabia's online marketplace will provide a variety of researchers and managers with will provide useful insights to various researchers and managers.
Author Response
Dear reviewer 2, please see attached file.

Reviewer 3 Report
In this paper, the authors investigated the effect of artificial intelligence on the end-user purchasing intentions of Saudi Arabian customers. The research topic is meaningful and interesting, but there are some flaws in the paper that can be addressed by the authors.
1. Motivations and contributions of this study should be highlighted in the introduction section.
2. It is suggested to discuss managerial implications of the results in the Discussion section.
3. Sections 7 and 8 can be combined and reduced.
4. The references can be improved by adding some new and closed related researches.
5. The manuscript needs a throughout revision concerning the language.
Author Response
Dear reviewer 3, please see attached file.

Round 2
Reviewer 1 Report
I am happy to see that the authors have tried their best to address reviewers' comments. Still, use of language is problematic. Other than that, I have no further comments on the paper.
Author Response
Dear reviewer 1,
Many thanks for your comment. Please note that the document went through extensive editing by MDPI order number (english-48196) and I will upload their certificate into the system.
Regards,
Hasan
